# Adoption of Hypofractionated and Ultrahypofractionated Adjuvant Radiation Therapy for Breast Cancer Across Main and Community Centers Within a Single Healthcare System

**DOI:** 10.3390/curroncol32110619

**Published:** 2025-11-06

**Authors:** Leila T. Tchelebi, Ajay Kapur, Clary Evans

**Affiliations:** Northwell, New Hyde Park, NY 11040, USA

**Keywords:** breast cancer, adjuvant radiation, fractionation

## Abstract

After breast cancer surgery, many people receive radiation therapy, which can be delivered over many or few visits. We asked whether doctors at our main university hospital switched to shorter schedules faster than doctors at our eight community clinics. We reviewed all patients treated from 2017 to 2022 and grouped care into long courses (about 25 visits), short courses (about 15), and very short courses (5). We compared care before 2020 and after 2020 for treatment to the breast and to the chest area after breast removal surgery. Long courses became less common everywhere over time. By 2020–2022, the main hospital used long courses less often and short or very short courses more often than the community clinics. Shorter care can reduce time, travel, and cost for patients. A shared treatment guide could help make care more consistent across all sites.

## 1. Introduction

More than three hundred thousand women are diagnosed with breast cancer in the United States each year, and the majority of patients who have definitive surgery receive adjuvant radiation [1]. Adjuvant breast radiation has evolved over the last few decades as clinical trials have demonstrated the safety and efficacy of shorter courses of treatment, which are less onerous on cancer patients [2,3,4,5,6,7,8]. In 2011, the use of hypofractionation for breast cancer patients was endorsed by the American Society for Radiation Oncology [9], and in 2015, the first recommendation in “Choosing Wisely: The American Society for Radiation Oncology’ Top 5 List” was to consider shorter treatment schedules when planning to administer whole breast radiation to patients [10].

Despite the preponderance of prospective data in support of shorter fractionation for breast cancer, adoption of hypofractionation has been relatively slow and non-uniform across treatment centers [11,12,13]. While the reasons for this variation are multi-factorial, primary reasons likely include individual practioners’ preference and familiarity with certain treatment regimens [14], income-based pressures since longer treatment regimens are more favorably compensated, and variations in reimbursement across health systems [15]. Studies have shown that differences in management of breast cancer patients exist between providers at academic practices versus community centers in the use of certain radiation treatment techniques [16]. However, little is known with regard to the use of breast fractionation regimens and adoption of hypofractionation by physicians practicing at main academic centers in comparison with physicians practicing at community centers [13].

The COVID-19 pandemic in 2020 resulted in a significant push within radiation medicine departments to adopt hypofractionated treatment regimens to minimize exposure to the virus [17]. The present paper sought to analyze practice patterns within our large healthcare network and compare the adoption of hypofractionated (including ultra-hypofractionated) radiation for breast cancer patients before and after 2020 between the main academic center and community sites. We hypothesized that the global drive to hypofractionate during the COVID-19 pandemic would mitigate any potential differences in fractionation patterns between the academic and community centers.

## 2. Materials and Methods

Starting in 2007, our Department of Radiation Medicine began the process of developing consensus-based treatment pathways based on the Institute of Medicine outline for guideline development [18]. These directives are intended to standardize clinical practice in our geographically complex modern healthcare system. The directives provide guidance on radiation dose, fractionation, and technique, set-up for CT simulation, imaging acquisition during treatment, and guidelines for contouring and planning, including dose constraints. The directives are created by members of the directives committee, which consists of volunteer dosimetrists, physicists, and physicians (from either the main center or community sites) who are tasked with reviewing and updating our directives based on the latest practice-changing published protocols. The committee meets weekly to ensure that directives are updated in a timely manner to reflect the current standard of care treatment in the management of cancer patients. When there is disagreement on best practices, decisions are reached via consensus vote. Following the publication of several prospective randomized trials for breast hypofractionation and ultra-fractionation, a set of breast cancer directives reflecting these modern treatment regimens were developed. To assess compliance with these directives, we assessed their use across our department.

All patients treated with adjuvant breast and chest wall RT between 2017 and 2022 in our radiation oncology department were identified from our treatment planning database. Patients who received palliative radiation for breast cancer or a non-directive dose were excluded. Patients who whose bilateral breast or chest wall were treated were counted as receiving two separate treatments. Boost or supraclavicular fields were not counted as separate treatments. Using our treatment directives, various treatment techniques were identified and evaluated. Standard fractionation consisted of treatment delivered in 25–28 fractions to a total dose of 50–50.4 Gy. Moderate hypofractionation consisted of treatment delivered in 15–16 fractions to a total dose of 40.05–42.56 Gy. Finally, ultra hypofractionation consisted of treatment delivered in 5 fractions to a total dose of 26–30 Gy. These dose/fractionation schemes could be delivered to either the whole breast, partial breast, or chest wall regions. Moderate hypofractionation and ultra-hypofractionation for both intact whole and partial breast were also summed and reported in aggregate as “Breast: hypofractionation.” Similarly, the use of hypofractionation and ultra-hypofractionation for chest wall patients were summed and reported in aggregate as “Chest Wall: Hypofractionation.”

Our large health system consists of one main academic center (“Main”) and community sites (“Community”) distributed over a wide geographic region within a single State. Patients who were treated at the main site were placed in the “Main Site” group and patients who were not treated in the main site were placed in the “Community” group. Use of each technique was compared between the Main versus Community sites in two time periods preceding and following the onset of the COVID-19 pandemic, 2017–2019 and 2020–2022, respectively. Differences were assessed using z-ratios for the difference between independent proportions.

## 3. Results

### 3.1. Overall Cohort

A total of 3780 breast cancer treatments were included: 1643 (43%) were treated at the main campus and 2137 (57%) patients were treated at community sites. Patients were treated at one main site (CFAM) and eight community sites (ICC, LHH, NSRT, NWH, PMH, QRC, SIUH and LIJ). Table 1 shows the overall breast cancer patient cohort treated at both the main center and the community sites in the early (2017–2019) and late (2020–2022) periods broken down by treatment site (whole breast, partial breast and chest wall) and by fractionation (standard fractionation, hypofractionation and ultra-hypofractionation)). This is also illustrated in Figure 1. 

### 3.2. Main Campus

There was a statistically significant decrease in the use of standard fractionation for intact breast patients at the main center from the early period to the late period (11.7% of all breast patients being treated at the main center in the early period versus 2.0% in the later period, *p* < 0.01) (Table 2). There was a statistically significant decrease in the use of standard fractionation for chest wall patients at the main center between the early and the late period (96.6% early versus 37.4% late, *p* < 0.01). There was also a statistically significant increase in the use of hypofractionation for chest wall patients at the main center from the early period to the late period (3.4% baseline versus 62.6% follow-up, *p* < 0.01).

### 3.3. Community

There was a statistically significant decrease in use of standard fractionation at community centers from the early to the late period (11.6% early versus 7.8.% late, *p* = 0.01) (Table 3). There was a statistically significant decrease in the use of standard fractionation for chest wall patients in the community between the early and the late periods (97.4% early versus 80.7% late, respectively, *p* < 0.01). There was also a significant increase in the use of hypofractionation at community centers for patients undergoing treatment to the chest wall from the early to the late period (2.6% early versus 19.3% late, respectively, *p* < 0.01).

### 3.4. Main Versus Community

There was no difference in the use of standard fractionation for intact breast cancer patients between the main center and community centers for the time period overall (7.4% main versus 8.8% community, *p* = 0.15) (Table 4) or during the early period (11.7% main versus 11.6% community, *p* = 0.95). However, more patients were treated with standard fractionation at the community centers versus the main center during the late period (2.0% for main versus 7.8% for community, *p* < 0.01). Meanwhile, more patients at the main center were treated with hypofractionation relative to the community sites in the late period (98.0% main versus 92.2% community, *p* < 0.01).

For chest wall patients, standard fractionation was used more frequently at community sites versus the main site for the period overall (69.9% main versus 84.8% community, *p* < 0.01). There was no difference during the early period, but there was an increased use of standard fractionation during the late period at community sites versus the main center (80.7% community versus 37.4% main, *p* < 0.01). There was a statistically higher use of hypofractionation for chest wall patients at the main center versus the community centers for the time period overall (30.1% main versus 15.2% community, *p* < 0.01). Use of chest wall hypofractionation did not differ during the early period, but there was an increased use at the main center versus community sites in the late period (19.3% community versus 62.6% main, *p* < 0.01).

**Figure 1 curroncol-32-00619-f001:**
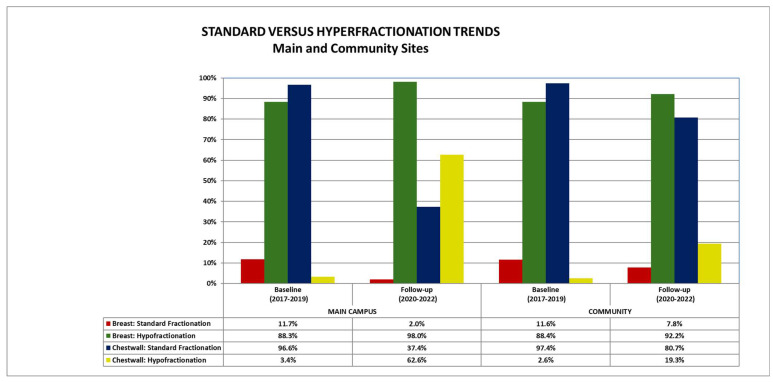
Standard versus hypofractionation trends over time at the main and community sites.

## 4. Discussion

To our knowledge, this is the first study to describe the adoption of both hypofractionated and ultrahypofractionated adjuvant radiation therapy for breast cancer treatment among providers at a large academic medical center. We found that community sites were more likely to continue to use standard fractionation for the treatment of intact breast and chest wall radiation, as compared with the main site, during the later time period and community centers were less likely to adopt shorter treatment schedules for both intact and chest wall breast cancer patients, as compared to the main center, during the later time period. The difference was particularly notable in the adoption of hypofractionation for chest wall patients with only 19% of cases in the community versus 63% at the main center using this shorter regimen in 2020 and beyond.

These results highlight the heterogenicity of practice even within a single large healthcare institution, which utilizes shared treatment protocols and standard operating practices. Standardizing cancer care among treatment centers is important for delivering high-quality care to patients. The disparate adoption of shorter fractionation schemes at our community sites relative to our main center highlights the need to implement measures to enhance standardization of patient care across healthcare networks to ensure that the same quality of care is delivered everywhere.

High-quality data to support the use of shorter fractionation regimens for breast cancer patients was first published in 2006 by Owen et al. who randomly assigned 1410 women in the United Kingdom (UK) with invasive breast cancer to receive either 50 Gy in 25 fractions, 39 Gy in 13 fractions, or 42.9 Gy in 13 fractions, and found no difference in ipsilateral tumor relapse at 10 years based on dose/fractionation [19]. This finding was replicated in the UK START B trial in 2008 in which 2215 women with invasive breast cancer were randomized to receive either 50 Gy in 25 fractions of 2.0 Gy over 5 weeks or 40 Gy in 15 fractions of 2.67 Gy over 3 weeks. There was no difference in local control rates at 10 years and the hypofractionated schedule was associated with a lower rate of adverse effects [6]. Since the publication of these landmark trials, several other large randomized controlled studies have demonstrated the safety and efficacy of hypofractionation in the adjuvant treatment of intact breast cancer patients [4,5].

Randomized data supporting the use of hypofractionation for post-mastectomy chest wall patients is more scarce, with the first being published in 2019 in China [7]. Wang et al. randomized 820 patients who had undergone mastectomy to receive chest wall and nodal irradiation at a dose of 50 Gy in 25 fractions over 5 weeks or 43.5 Gy in 15 fractions over 3 weeks [7]. They found that post-mastectomy hypofractionated radiotherapy was non-inferior to and had similar toxicities to conventional fractionated radiotherapy in patients with high-risk breast cancer after mastectomy. The HypoG-01 phase III clinical trial demonstrated the safety of hypofractionation to nodal basins, whether in the post-mastectomy or breast-conserving setting [20]. Finally, results of the randomized trial of hypofractionated post-mastectomy radiation therapy in women with breast reconstruction (RT-CHARM) further support the safety of hypofractionated radiation following mastectomy and reconstruction [21].

More recently, high-quality randomized data supporting the use of ultra-hypofractionation has been published. FAST-Forward, published in 2020, was a multicenter, phase 3, randomized, non-inferiority trial conducted in the UK which compared 40 Gy in 15 fractions (over 3 weeks) to 27 Gy in five fractions (over 1 week), to 26 Gy in five fractions (over 1 week), to the whole breast or chest wall [3]. Of note, lymph nodes were not treated in chest wall cases. The authors found that 26 Gy in 5 fractions was non-inferior to the hypofractionated 3-week regimen with similar toxicity rates. The UK FAST trial, also published in 2020, randomized intact breast cancer patients to receive 50 Gy in 25 fx over 5 weeks or 30 or 28.5 Gy in 5 once-weekly fx of 6.0 or 5.7 Gy [2]. The primary endpoint was a change in photographic breast appearance with local control as a secondary endpoint. At 10 years, there was no significant difference in normal tissue effect rates after 28.5 Gy in 5 fx compared with 50 Gy in 25 fx, but normal tissue effects were higher after 30 Gy in 5 fx.

As a result of the preponderance of high-quality evidence showing the safety and efficacy of shorter fractionation schemes for breast cancer patients, the 2015 publication of “Choosing Wisely: The American Society for Radiation Oncology’ Top 5 List” recommended consideration of shorter treatment schedules when planning to administer whole breast radiation to patients [10]. Despite these data and this recommendation, studies have shown variability in the adoption of shorter fractionation courses for breast cancer patients. Bekelman et al. found that the minority of patients who met criteria for treatment with hypofractionation were actually treated with hypofractionation [22]. A study published by Corrigan et al. found that physician variation accounted for 21.7% of variance in the adoption of ultra-hypofractionation for intact breast cancer patients, with rate of use by the treating physicians ranging from 0% to 75.6% [13]. In another study evaluating the geographic heterogeneity in the delivery of hypofractionated radiation therapy for breast cancer among Medicare beneficiaries across the United States found that the proportion of women receiving hypofractionated breast radiation in individual hospital referral regions varied from 0% to 61% [23].

There are several reasons that may account for the slower adoption of hypofractionation at our community sites versus our main center. One of these may be the availability of clinical trials testing shorter fractionation schemes, which were open at the main center and not at the community sites. The Alliance A221505 (RT CHARM) [21] trial which compared patients following mastectomy and reconstruction to standard radiation versus hypofractionation was open at our main center during this time but not at our community sites. Data has shown that practitioners required to adopt new radiotherapy techniques for the purposes of clinical trial enrollment are more likely to implement these novel techniques when the trial data is published in their support [24]. In general, providers at community sites who do not have a site-specific treatment focus may be less comfortable adopting newer treatments than providers at the main center with a focus on breast cancer who treat a much higher volume of these patients and thus may feel more comfortable adopting newer treatment paradigms. Financial incentives may also have motivated longer fractionation schemes at community sites. As with many radiation therapy departments, bonus payments/incentive compensation often involves a Relative Value Unit (RVU) component in which physicians are paid more if they accrue more RVUs. Longer treatment courses, as with standard fractionation, result in more RVUs than shorter treatment courses, which may result in increased physician payments, thus motivating community site providers to offer more protracted treatments [25]. Concern has been raised that the current fee-for-service payment system may slow the adoption of some evidence-based practices, such as hypofractionation [25].

The COVID-19 pandemic in 2020 posed a severe threat to cancer patients requiring to present for several weeks of daily treatment with potential exposure to the virus at each visit. As a result, there was a systematic push across cancer disease sites to use hypofractionated treatment regimens [26]. For our study, we thus elected to analyze the adoption of shorter courses of radiation for breast cancer patients in the time period before and after 2020, hypothesizing that this push for hypofractionation during the pandemic would result in uniform adoption of hypofractionation across the eight radiation treatment facilities in our large health network. We found, however, that our community centers did not adopt shorter fractionation schemes in the wake of the pandemic relative to our main center.

There are a number of limitations to this study including the retrospective nature of our analysis. One limitation is the study period includes a period of time in which the healthcare system underwent a large expansion, incorporating a number of new small community centers at the end of the “Early” period. Accordingly, the number of patients treated in the community sites substantially increased at the follow-up period compared with the baseline. Standardizing radiotherapy practices remains a challenge in the evolving U.S. healthcare market, where ongoing acquisitions and mergers continuously reshape clinical environments. These structural changes can inadvertently reintroduce variations in practice, even within a single healthcare system. We acknowledge that this may be a factor in our own institution, which expanded during the study period through the acquisition of community centers. As a result, differences in radiation therapy adoption may reflect not only provider preferences but also the integration of diverse clinical cultures and protocols. Practice guidelines, departmental directives and peer-review can help standardize practice to ensure all patients receive optimal and appropriate adjuvant radiation for breast cancer.

## 5. Conclusions

The present study shows that despite recent trial evidence supporting the use of shorter radiation treatments for both intact breast and chest wall breast cancer patients, our community sites less readily adapted to the recommended changes in practice versus our main academic center. This relative reluctance to adopt hypofractionation as readily at our community centers despite the use of centralized mandated protocols highlights a failure to standardize practice even within a single healthcare system, which may adversely impact patient care. The reasons for this difference are not known; however, standardization of treatment by implementation of an adjuvant radiation treatment algorithm may facilitate uniform care among patients with breast cancer and we are investigating the impact of this approach.

## Figures and Tables

**Table 1 curroncol-32-00619-t001:** Overall breast cancer patient cohort.

PATIENTS	MAIN CAMPUS			COMMUNITY			TOTAL
Number of Patients Treated	Early (2017–2019)	Late (2020–2022)	Total	Early (2017–2019)	Late (2020–2022)	Total	Total
**Breast: Standard**	89	12	101	53	107	160	261
**Breast: Hypofractionation**	669	524	1193	403	1156	1558	2751
**Breast: Ultra Hypofractionation**	0	49	49	0	45	45	94
**Breast, Partial: Hypofractionation**	0	0	0	0	2	2	2
**Breast, Partial: Ultra Hypofractionation**	0	29	29	0	58	58	87
**Breast Subtotals**	758	614	1371	456	1368	1823	3194
**Chest Wall: Standard**	144	46	190	75	192	267	457
**Chest Wall: Hypofractionation**	5	77	82	2	43	45	127
**Chest Wall: Ultra Hypofractionation**	0	0	0	0	3	3	3
**Chest Wall Subtotals**	149	123	272	77	238	314	586
**Grand Total**	907	737	1643	533	1606	2137	3780

**Table 2 curroncol-32-00619-t002:** Breast cancer patients treated at main campus in the early and late periods.

PATIENTS	MAIN CAMPUS		
Percentage of Patients Treated at the Main Campus	Early (2017–2019)	Late (2020–2022)	*p*-Value (Early vs. Late)
**Intact Breast**			
**Breast: Standard fractionation**	11.7%	2.0%	**<0.01**
**Breast: Hypofractionation ***	88.3%	98.0%	**<0.01**
Breast, Whole: Moderate hypofractionation	88.3%	85.3%	0.12
Breast, Whole: Ultra hypofractionation	0.0%	8.0%	NA
Breast, Partial: Moderate hypofractionation	0.0%	0.0%	NA
Breast, Partial: Ultra hypofractionation	0.0%	4.7%	NA
**Chest Wall**			
**Chest Wall: Standard fractionation**	96.6%	37.4%	**<0.01**
**Chest Wall: Hypofractionation ^**	3.4%	62.6%	**<0.01**
Chest Wall: Moderate hypofractionation	3.4%	62.6%	**<0.01**
Chest Wall: Ultra hypofractionation	0.0%	0.0%	NA

* This includes moderate hypofractionation and ultra hypofractionation to whole and partial breast. ^ This includes moderate hypofractionation and ultra hypofractionation to the chest wall. Bolded text indicates statistically significant differemces.

**Table 3 curroncol-32-00619-t003:** Breast cancer patients treated at community sites in the early and late periods.

PATIENTS	COMMUNITY		
Percentage of Patients Treated at Community	Early (2017–2019)	Late (2020–2022)	*p*-Value (Early vs. Late)
**Intact Breast**			
**Breast: Standard fractionation**	11.6%	7.8%	**0.01**
**Breast** **:** **Hypofractionation ***	88.4%	92.2%	**0.01**
Breast, Whole: Moderate Hypofractionation	88.4%	84.5%	0.04
Breast, Whole: Ultra Hypofractionation	0.0%	3.3%	NA
Breast, Partial: Moderate Hypofractionation	0.0%	0.1%	NA
Breast, Partial: Ultra Hypofractionation	0.0%	4.2%	NA
**Chest Wall**			
**Chest Wall: Standard fractionation**	97.4%	80.7%	**<0.01**
**Chest Wall: Hypofractionation ^**	2.6%	19.3%	**<0.01**
**Chest Wall: Moderate Hypofractionation**	2.6%	18.1%	<0.01
Chest Wall: Ultra Hypofractionation	0.0%	1.3%	NA

* This includes moderate hypofractionation and ultra hypofractionation to whole and partial breast. ^ This includes moderate hypofractionation and ultra hypofractionation to the chest wall.

**Table 4 curroncol-32-00619-t004:** Breast cancer patients treated at the main campus versus the community sites in the early and late periods.

PATIENTS	Main Versus Community								
Time Period	Early (2017–2019)			Late (2020–2022)			Whole Period (2017–2022)		
Fractionation Scheme	Main	Community	*p*-Value	Main	Community	*p*-Value	Main	Community	*p*-Value
**Intact Breast**									
**Breast: Standard fractionation**	11.7%	11.6%	**0.95**	2.0%	7.8%	**<0.01**	7.4%	8.8%	**0.15**
**Breast:** **Hypofractionation ***	88.3%	88.4%	**0.95**	98.0%	92.2%	**<0.01**	92.6%	91.2%	**0.15**
Breast, Whole: Moderate Hypofractionation	88.3%	88.4%	0.95	85.3%	84.5%	0.63	87.0%	85.5%	0.21
Breast, Whole: Ultra Hypofractionation	0.0%	0.0%	NA	8.0%	3.3%	<0.01	3.6%	2.5%	0.07
Breast, Partial: Moderate Hypofractionation	0.0%	0.0%	NA	0.0%	0.1%	NA	0.0%	0.1%	NA
Breast, Partial: Ultra Hypofractionation	0.0%	0.0%	NA	4.7%	4.2%	NA	2.1%	3.2%	NA
**Chest Wall**									
**Chest Wall: Standard Fractionation**	96.6%	97.4%	**0.76**	37.4%	80.7%	**<0.01**	69.9%	84.8%	**<0.01**
**Hypofractionation ^**	3.4%	2.6%	**0.76**	62.6%	19.3%	**<0.01**	30.1%	15.2%	**<0.01**
Chest Wall: Moderate Hypofractionation	3.4%	2.6%	NA	62.6%	18.1%	<0.01	30.1%	14.3%	<0.01
Chest Wall: Ultra Hypofractionation	0.0%	0.0%	NA	0.0%	1.3%	NA	0.0%	1.0%	NA

* This includes moderate hypofractionation and ultra hypofractionation to whole and partial breast. ^ This includes moderate hypofractionation and ultra hypofractionation to the chest wall.

## Data Availability

The original contributions presented in this study are included in the article. Further inquiries can be directed to the corresponding author.

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
