# Peer review of "Adoption of Hypofractionated and Ultrahypofractionated Adjuvant Radiation Therapy for Breast Cancer Across Main and Community Centers Within a Single Healthcare System"

_curroncol, 2025, doi:10.3390/curroncol32110619_

Round 1

Reviewer 1 Report

Comments and Suggestions for Authors

General comments

  • This study highlights how differences in clinical setup and practitioner preferences contribute to variability in the treatment of breast cancer patients receiving fractionated radiotherapy before and after the COVID-19 pandemic. These findings are valuable for regulatory bodies and professional associations, as they can inform the development of standardized treatment guidelines to promote consistent care delivery, regardless of individual practitioner approaches.

Major comment

  • Although this study utilized retrospectively collected data from patients previously treated according to standard-of-care protocols, it involved the systematic investigation of patient outcomes with the intent to contribute to generalizable knowledge. Therefore, Institutional Review Board (IRB) approval was obtained to ensure compliance with ethical standards and regulatory requirements for research involving human subjects.
  • Include the inclusion and exclusion criteria, if any, or mention in the method section how the patients are screened and included in the study.

Minor comment

  • Figure 1: The color and the values in each bar graph do not match; use better-matched text-graph color.

Author Response

Reviewer 1

General comments

  • This study highlights how differences in clinical setup and practitioner preferences contribute to variability in the treatment of breast cancer patients receiving fractionated radiotherapy before and after the COVID-19 pandemic. These findings are valuable for regulatory bodies and professional associations, as they can inform the development of standardized treatment guidelines to promote consistent care delivery, regardless of individual practitioner approaches.

Response: Thank you for providing these comments.

Major comment

  • Although this study utilized retrospectively collected data from patients previously treated according to standard-of-care protocols, it involved the systematic investigation of patient outcomes with the intent to contribute to generalizable knowledge. Therefore, Institutional Review Board (IRB) approval was obtained to ensure compliance with ethical standards and regulatory requirements for research involving human subjects.

Response: Thank you for this helpful comment. This study was conducted as part of a quality improvement (QI) project within our radiation medicine department to improve standardization across our network. As a QI initiative, it was considered exempt from IRB regulation.

  • Include the inclusion and exclusion criteria, if any, or mention in the method section how the patients are screened and included in the study.

Response: Thank you for this comment. We have added inclusion/exclusion criteria to the second paragraph of the Methods section.

Minor comment

  • Figure 1: The color and the values in each bar graph do not match; use better-matched text-graph color.

Response: Thank you for this suggestion. We have modified the colors in Figure 1 and moved the percentages to a table below for clarity.

Reviewer 2 Report

Comments and Suggestions for Authors

The present article examines whether or not physicians at main academic centers adopt hypofractionated regimens for adjuvant breast cancer radiation more readily than those working at community centers. The conclusion is that hypofractionation was more rapidly introduced  at  the main academic center as compared to community sites. Putative explanation is discussed.

Comments:

Redundance. There is a lot of repetitive numerical data in Text and Tables and Figure 1.

Reference. Add a reference for the 70% adjuvant RT (L40-42).

Finance. Do provide details about the financial considerations (L60).  The discussion (L234 – 242) of this point is confusing. What is the cost of hypofractionated as compared to standard treatment? Provide numerical details about RVUs (explain abbreviation) for the cohorts under consideration (L257).

Selection. How were patients identified (L87)? Was selection applied to reach the numbers displayed in Table 1? The grand total for total community is much higher than for total main campus, in contrast with grand totals for early. See also L271- 274, and 279: were the 8 communities not the same during the whole study period? What is the total volume of breast cancer patients in both cohorts? See also L 254. Could the authors provide details, name and location, about the participating centers, one academic and 8 community (L99) and define the “unified healthcare system” cited on L290.

Terminology. L 94-96 states that both “intact” and “partial breast” were summed and reported in aggregate as “Intact Breast”. Is A + B = A? L109 and Table 1  breaks down by “breast” and “partial breast”. Figure 1 and Table 2 say  “breast”. L 114 introduces a new group, “reconstructed breast”, not mentioned in the Materials section. Uniform terminology is needed.

Author Response

Reviewer 2

The present article examines whether or not physicians at main academic centers adopt hypofractionated regimens for adjuvant breast cancer radiation more readily than those working at community centers. The conclusion is that hypofractionation was more rapidly introduced  at  the main academic center as compared to community sites. Putative explanation is discussed.

Comments:

Redundance. There is a lot of repetitive numerical data in Text and Tables and Figure 1.

Response: Thank you for this helpful comment. We have removed the redundancy from the text and only present in numerically in the tables/figure.

Reference. Add a reference for the 70% adjuvant RT (L40-42).

Response: Thank you for this comment. The sentence was revised and a reference provided.

Finance. Do provide details about the financial considerations (L60).  The discussion (L234 – 242) of this point is confusing. What is the cost of hypofractionated as compared to standard treatment? Provide numerical details about RVUs (explain abbreviation) for the cohorts under consideration (L257).

Response: Thank you for these helpful comments. Details regarding RVU compensation and its association to fractionation has been added.

Selection. How were patients identified (L87)? Was selection applied to reach the numbers displayed in Table 1? The grand total for total community is much higher than for total main campus, in contrast with grand totals for early. See also L271- 274, and 279: were the 8 communities not the same during the whole study period? What is the total volume of breast cancer patients in both cohorts? See also L 254. Could the authors provide details, name and location, about the participating centers, one academic and 8 community (L99) and define the “unified healthcare system” cited on L290.

Response: Thank you for these helpful comments. The community sites were the same but many had been acquired during the early years. These practices grew and matured during the early period which is reflected in the larger numbers in the later years as the practices were more established by the later time period. We have added this to the discussion.

Terminology. L 94-96 states that both “intact” and “partial breast” were summed and reported in aggregate as “Intact Breast”. Is A + B = A? L109 and Table 1  breaks down by “breast” and “partial breast”. Figure 1 and Table 2 say  “breast”. L 114 introduces a new group, “reconstructed breast”, not mentioned in the Materials section. Uniform terminology is needed.

Response: Thank you for these helpful comments. We have unified our terminology and redone the Figure.